# Identification of skill in an online game: The case of Fantasy Premier League

**Joseph D. O'Brien** [ID]*, **James P. Gleeson, David J. P. O'Sullivan**

MACSI, Department of Mathematics and Statistics, University of Limerick, Limerick, Ireland

* joseph.obrien@ul.ie

## Abstract

In all competitions where results are based upon an individual's performance the question of whether the outcome is a consequence of skill or luck arises. We explore this question through an analysis of a large dataset of approximately one million contestants playing *Fantasy Premier League*, an online fantasy sport where managers choose players from the English football (soccer) league. We show that managers' ranks over multiple seasons are correlated and we analyse the actions taken by managers to increase their likelihood of success. The prime factors in determining a manager's success are found to be long-term planning and consistently good decision-making in the face of the noisy contests upon which this game is based. Similarities between managers' decisions over time that result in the emergence of 'template' teams, suggesting a form of herding dynamics taking place within the game, are also observed. Taken together, these findings indicate common strategic considerations and consensus among successful managers on crucial decision points over an extended temporal period.

## Introduction

Hundreds of millions of people consume sporting content each week, motivated by several factors. These motivations include the fact that the spectator enjoys both the quality of sport on display and the feeling of eustress arising from the possibility of an upset [1, 2]. This suggests that there are two important elements present in sporting competition: a high level of skill among players that provides aesthetic satisfaction for the spectator and also an inherent randomness within the contests due to factors such as weather, injuries, and in particular luck. The desire for consumers to get further value from their spectating of sporting content has resulted in the emergence of *fantasy sports* [3–6], in which the consumers, or *managers* as we shall refer to them throughout this article, begin the season with a virtual budget from which to build a team of *players* who, as a result of partaking in the real physical games, receive points based upon their statistical performances. The relationship between the fantasy game and its physical counterpart raises the question of whether those who take part in the former suffer (or gain) from the same combination of skill and luck that makes their physical counterpart enjoyable.

University of Limerick, Department of Mathematics and Statistics at macsi@ul.ie.

**Funding:** This work was supported by Science Foundation Ireland grant numbers 16/IA/4470, 16/RC/3918, 12/RC/2289 P2 and 18/CRT/6049), co-funded by the European Regional Development Fund (J.P.G) URLs - https://www.sfi.ie/; https://ec.europa.eu/regional_policy/en/funding/erdf/; The funders had no role in study design, data collection and analysis, decision to publish, or preparation of the manuscript.

**Competing interests:** The authors have declared that no competing interests exist.

The emergence of large scale quantities of detailed data describing the dynamics of sporting games has opened up new opportunities for quantitative analysis, both from a team perspective [7–14] and also at an individual level [15–21]. This has resulted in analyses aiming to determine two elements within the individual sports; firstly quantifying the level of skill in comparison to luck in these games [9, 22–25] while, secondly, identifying characteristics that suggest a difference in skill levels among the competing athletes [17, 26]. Such detailed quantitative analysis is not, however, present in the realm of fantasy sports, despite their burgeoning popularity with an estimated 45.9 million players in the United States alone in 2019 [27]. Two notable exceptions however are recent studies which demonstrate quantitatively that skill was a more important factor than luck in fantasy sports based upon American sports, analytically in the case of [28] and via a detailed statistical analysis in [29].

Motivated by this body of work, we consider a dataset describing the *Fantasy Premier League (FPL)* [30], which is the online fantasy game based upon the top division of England's football league. This game consists of over seven million *managers*, each of whom builds a virtual team based upon real-life players. Before proceeding, we introduce a brief summary of the rules underlying the game, to the level required to comprehend the following analysis [31]. The (physical) Premier League consists of 20 teams, each of whom play each other twice, resulting in a season of 380 fixtures split into 38 unique *gameweeks*, with each gameweek generally containing ten fixtures. A manager in FPL has a virtual budget of £100m at the initiation of the season from which they must build a squad of 15 players from the approximately 600 available. Each player's price is set initially by the game's developers based upon their perceived value to the manager within the game, rather than their real-life transfer value. The squad of 15 players is composed under a highly constrained set of restrictions which are detailed in S1 Note in S1 File.

In each gameweek the manager must choose 11 players from their squad as their team for that week and is awarded a points total from the sum of the performances of these players (see S1 Table in S1 File). The manager also designates a single player of the 11 to be the captain, with the manager receiving double this players' points total in that week. Between consecutive gameweeks the manager may also make one unpenalised change to their team, with additional changes coming as a deduction in their points total. The price of a given player then fluctuates as a result of the supply-and-demand dynamic arising from the transfers across all managers' rosters. The intricate rules present multiple decisions to the manager and also encourages longer-term strategising that factors in team value, player potential, and many other elements.

With these complexities in mind, in this article we conduct a thorough statistical analysis of the performance of managers within the game to provide evidence of skill being a strong factor in the level of success obtained by competitors. We begin by analysing the historical performance of managers in terms of where they have ranked within the competition alongside their points totals in multiple seasons, in some cases over a time interval of up to thirteen years. We find a consistent level of correlation between managers' performances over seasons, suggesting a persistent level of skill over an extended temporal scale. Taking this as our starting point, we aim to understand the decisions taken by managers which are indicative of this skill level over the shorter temporal period of the 38 gameweeks making up the 2018/19 season by analysing the entire dataset of actions taken by the majority of the top one million managers (Due to data availability issues at the time of collection such as managers not taking part in the entire season, the final number of managers identified was actually 901,912. We will however, for the sake of brevity, refer to these as the top 1 million managers over the course of this article. It is also important to note that data from previous seasons is unattainable, which is why we restrict this detailed study to the 2018/19 season.) over the course of the season. Even at this shorter

scale we find consistent tiers of managers who, on a persistent basis, outperform those at a lower tier.

With the aim of identifying why these differences occur, we present evidence of consistently good decision making with regard to team selection and strategy. This would be consistent with some common form of information providing these skilled managers with an 'edge' of sorts, for example in the US it has been suggested that 30% of fantasy sports participants take advantage of further websites when building their teams [32]. Arguably most interesting of all, we demonstrate how at points throughout the season there occurs temporary herding behaviour in the sense that managers appear to converge to consensus on a *template team*. However, the consensus does not persist in time, with managers subsequently differentiating themselves from the others. We consider possible reasons and mechanisms for the emergence of these template teams.

## Results

### Historical performance of managers

We consider two measures of a manager's performance in a given season of FPL: the total number of points their team has obtained over the season and also their resulting rank based on this points total in comparison to all other managers. A strong relationship between the managers' performances over multiple seasons of the game is observed. For example, in panel (a) of Fig 1 we compare the ranks of managers who competed in both the 2018-19 and 2017-18 seasons. The density near the diagonal of this plot suggests a correlation between performances in consecutive seasons. Furthermore, we highlight specifically the bottom left corner which indicates that those managers who are among the most highly ranked appear to perform well in both seasons. Importantly, if we consider the top left corner of this plot it can be readily seen that the highest performing managers in the 2017-18 season, in a considerable number of cases, did not finish within the lowest positions in the following season as demonstrated by the speckled bins with no observations.

This is further corroborated in panel (b), in which we show the pairwise Pearson correlation between the total points obtained by managers from seasons over a period of 12 years. While the number of managers who partook in two seasons tends to decrease with time, a

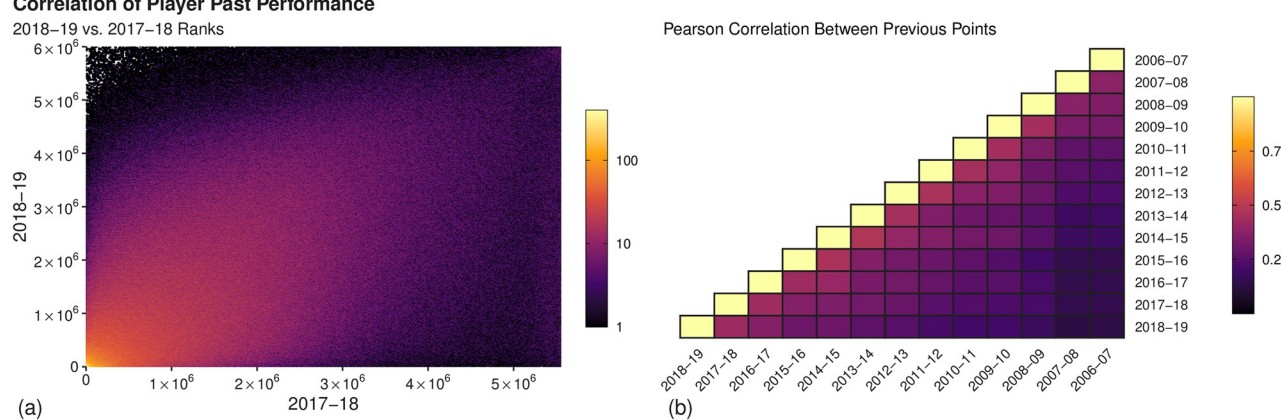

**Fig 1. Relationship between the performance of managers over seasons of FPL.** (a) The relationship between managers' ranks in the 2018/19 and 2017/18 seasons. Each bin is of width 5,000 with the colour highlighting the number of managers in each bin; note the logarithmic scale in colour. (b) The pairwise Pearson correlation between a manager's points totals over multiple seasons of the game, calculated over all managers who appeared in both seasons.

considerable number are present in each comparison. Between the two seasons shown in Fig 1(a) for example, we observe results for approximately three million managers and find a correlation of 0.42 among their points totals. Full results from 13 consecutive seasons, including the number managers present in each pair and the corresponding Pearson correlation coefficients, are given in S4 Table in S1 File.

Using a linear regression fit to the total points scored in the 2018/19 season as a function of the number of previous seasons in which the manager has played (S5 Table in S1 File) we find that each additional year of experience is worth on average 22.1 ($R^2$ = 0.082) additional points (the overall winner in this season obtained 2659 points). This analysis suggests that while there are fluctuations present in a manager's performance during each season of the game, there is also some consistency in terms of performance levels, suggesting a combination of luck and skill being present in fantasy sports just as was observed in their physical analogue in [28].

## Focus on season 2018-19

In the analysis above we considered, over multiple seasons, the performance of managers at a season level in terms of their cumulative performance over the 38 gameweeks of each season. We now focus at a finer time resolution, to consider the actions of managers at the gameweek level for the single season 2018/19, in order to identify elements of their decision making which determined their overall performance in the game.

The average points earned by all managers throughout the season is shown in Fig 2(a) along with the 95 inter-percentile range, i.e., the values between which the managers ranked in quantiles 0.025 to 0.975 appear. This quantity exhibits more frequent fluctuations about its long-term average (57.05 points per gameweek) in the later stages of the season, suggesting that some elements of this stage of the season cause different behaviour in these gameweeks. There may of course be many reasons for this e.g., difficult fixtures or injuries for generally high-scoring players or even simply a low/high scoring gameweek, which are themselves factors of luck within the sport itself (see S2 Table in S1 File for a detailed break down of points per gameweek). However, in the analysis to follow we consider an important driver of the fluctuations related to strategic decisions of managers in these gameweeks.

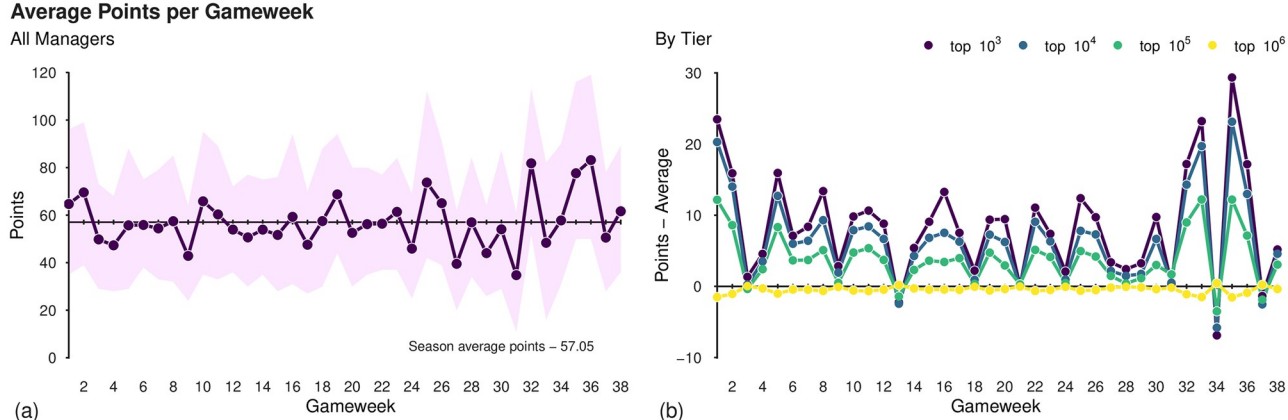

**Fig 2. Summary of points obtained by managers over the course of the 2018/19 season.** (a) The mean number of points over all managers for each GW. The shaded regions denote the 95% percentiles of the points' distribution. (b) The difference between the average number of points for four disjoint tiers of manager, the top $10^3$, $10^4$, $10^5$, and $10^6$, and the overall average points as per panel (a). Note that managers are considered to be in only one tier so, for example, the top-$10^4$ tier contains managers ranked from 1001 to $10^4$.

In each season some fixtures must be rescheduled due to a number of reasons, e.g., clashing fixtures in European competitions, which results in certain gameweeks that lack some of the complete set of ten fixtures. Such scenarios are known as *blank-gameweeks* (BGW) and their fixtures are rescheduled to another gameweek in which some teams play twice; these are known as *double-gameweeks* (DGWs). In the case of the 2018/19 season these BGWs took place in GWs 27 (where there were eight fixtures), 31 (five fixtures), and 33 (six fixtures), making it difficult for some managers to have 11 starting players in their team. The DGWs feature some clubs with two games and therefore players in a manager's team who feature in these weeks will have twice the opportunity for points; in the 2018/19 season these took place in GWs 25 (where 11 games were played), 32 (15), 34 (11), and 35 (14). We see that the main swings in the average number of points are actually occurring in these gameweeks (aside from the last peak in GW 36 which we will comment on later in the article). We will later show that the managers' attitude and preparation towards these gameweeks are in fact indicators of their skill and ability as a fantasy manager.

To analyse the impact of decision-making upon final ranks, we define *tiers* of managers by rank-ordering them by their final scores and then splitting into the top $10^3$, top $10^4$, top $10^5$, and top $10^6$ positions. These disjoint tiers of managers, i.e., the top $10^3$ is the managers with ranks between 1 and 1000, the top $10^4$ those with ranks between 1001 and 10,000 and so on, range from the most successful (top $10^3$) to the relatively unsuccessful (top $10^6$) and so provide a basis for comparison (see S2 and S3 Tables in S1 File for summaries of points obtained by each tier). The average performance of the managers in each tier (relative to the baseline average over the entire dataset) are shown in panel (b) of Fig 1. Note that the points for the top $10^6$ tier are generally close to zero as the calculation of the baseline value is heavily dependent upon this large bulk of managers. A detailed summary of each tier's points total, along with visualisation of the distribution of points total may be found in S1 Fig and S1 Table in S1 File. It appears that the top tier managers outperform those in other tiers, not only in specific weeks but consistently throughout the season which results in the competition for places in this top tier more difficult to obtain as the season progresses (S2 Fig in S1 File). This is particularly noticeable in the first gameweek, where the top $10^3$ managers tended to perform very strongly, suggesting a high level of preparation (in terms of squad-building) prior to the physical league starting. We also comment that the largest gaps between the best tier and the worst tier occur not only in two of the special gameweeks (DGW 35 and BGW 33) but also in GW 1, which suggests that prior to the start of the season these managers have built a better-prepared team to take advantage of the underlying fixtures. We note however that all tiers show remarkably similar temporal variations in their points totals, in the sense that they all experience simultaneous peaks and troughs during the season. See S2 Table in S1 File for a full breakdown of these values alongside their variation for each gameweek.

Having identified both differences and similarities underlying the performance in terms of total points for different tiers of managers we now turn to analysis of the actions that have resulted in these dynamics.

## Decision-making

**Transfers.** The performance of a manager over the season may be viewed as the consequence of a sequence of decisions that the manager made at multiple points in time. These decisions include which players in their squad should feature in the starting team, the formation in which they should set up their team, and many more. In the following sections we consider multiple scenarios faced by managers and show that those who finished within a higher tier tended to consistently outperform those in lower tiers.

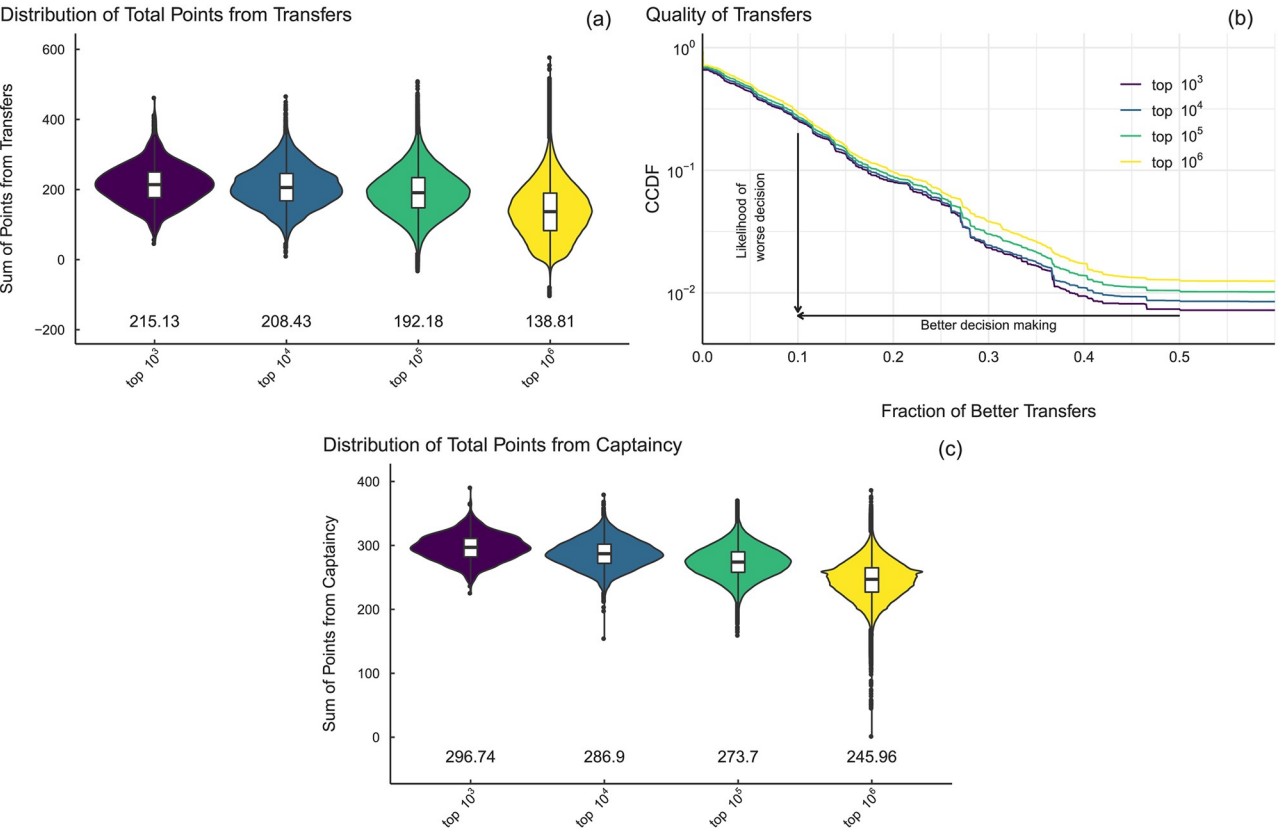

**Fig 3. Decisions of managers by tier.** (a) Distributions of the total net points earned by managers in the gameweek following a transfer, i.e., the points scored by the player brought in minus that of the player transferred out. The average net points for each tier is also shown below; note the difference between the top three tiers and the bottom tier. (b) Distribution of the fraction of better transfers a manager could have made based upon points scored in the following gameweek. Faster-decreasing distributions reflect managers in that tier being more successful with their transfers. (c) The distribution of points from captaincy along with the average total for each tier.

One decision the manager must make each gameweek is whether to change a player in their team by using a transfer. If the manager wants to make more than one transfer they may also do so but at the cost of a points deduction for each extra transfer. The distribution of total points made from transfers, which we determine by the difference between points attained by the player the manager brought in for the following gameweek compared to the player whom they transferred out, over the entire season for each tier is shown in Fig 3(a). The average number for each tier is also shown. To further analyse this scenario we calculate, for each gameweek, the number of better transfers the managers could have made with the benefit of perfect foresight, given the player they transferred out. This involves taking all players with a price less than or equal to that of the player transferred out and calculating the fraction of options which were better than the one selected, i.e., those who received more points the following gameweek (see Methods). Fig 3(b) shows the complementary cumulative distribution function (CCDF) of this quantity for each tier, note the steeper decrease of the CCDFs for the higher tiers implies that these managers were more likely to choose a strong candidate when replacing a player.

A second decision faced by managers in each gameweek is the choice of player to nominate as captain, which results in the manager receiving double points for this players' actions during the GW. This is, of course, a difficult question to answer as the points received by a player can be a function of both their own actions i.e., scoring or assisting a goal, and also their team's

collective performance (such as a defender's team not conceding a goal). This topic is an identification question which may be well suited to further research making use of the data describing the players and teams but with additional data about active managers who are making the same decision. For example, an analysis of the captaincy choice of managers based upon their social media activity was recently presented in [33] and showed that the *wisdom of crowds* concept performs comparably to that of the game's top managers. Panel (c) of Fig 3 shows the distribution of points obtained by managers in each tier from their captaincy picks. Again we observe that the distribution of points obtained over the season is generally larger for those managers in higher tiers.

**Financial cognizance.** The financial ecosystem underlying online games has been a focus of recent research [34, 35]. With this in mind, we consider the importance of managers' financial awareness in impacting their performance. As mentioned previously, each manager is initially given a budget of £100 million to build their team, constrained by the prices of the players which, themselves fluctuate over time. While the dynamics of player price changes occur via an undisclosed mechanism, attempts to understand this process within the community of Fantasy Premier League managers have resulted in numerous tools to help managers predict player price changes during the season, for example see [36]. The resulting algorithms are in general agreement that the driving force behind the changes is the supply and demand levels for players.

These price fluctuations offer an opportunity for the astute manager to 'play the market' and achieve a possible edge over their rivals and allow their budget to be more efficiently spent (see S4 Fig in S1 File for a description of player value and their corresponding points totals and S5 Fig in S1 File. for an indication of how the managers distribute their budget by player position). At a macro level this phenomenon of price changes is governed by the aforementioned supply and demand, but these forces are themselves governed by a number of factors affecting the player including, but not limited to, injuries, form, and future fixture difficulty. As such, managers who are well-informed on such aspects may profit from trading via what is in essence a fundamental analysis of players' values by having them in their team prior to the price rises [37]. Interestingly, we note that the general trend of team value is increasing over time among our managers as shown in panel (a) of Fig 4 along with corresponding 95 percentiles of the distribution, although there is an indicative decrease between weeks towards the season's end (GWs 31-35) suggesting the team value becomes less important to the managers towards the games conclusion. Equivalent plots for each tier are shown in S6 Fig in S1 File.

Probing further into the relationship between finance and the managers' rank, we show in Fig 4(b) the distribution of team values for the top two tiers (top $10^3$ and top $10^4$), compared with that for the bottom two tiers (top $10^5$ and top $10^6$) There is a clear divergence between the two groups from an early point in the season, indicating an immediate importance being placed upon the value of their team. A manager who has a rising team value is at an advantage relative to one who does not due to their increased purchasing power in the future transfer market. This can be seen in panel (c) of Fig 4 which shows the change in team value for managers at gameweek 19, the halfway point of the season, versus their final points total. A positive relationship appears to exist and this is validated by fitting an OLS Linear Regression with a slope of 21.8 ($R^2 = 0.1689$), i.e., an increase of team value by £1M at the halfway point is worth, on average, an additional 21.8 points by the end of the game (for the same analysis in other gameweeks see S5 Table in S1 File). The rather small $R^2$ value suggests, however, that the variation in a managers' final performance is not entirely explained by their team value and as such we proceed to analyse further factors which can play a part in their final ranking.

**Chip usage.** A further nuance to the rules of FPL is the presence of four *game-chips*, which are single use 'tricks' that may be used by a manager in any GW to increase their team's

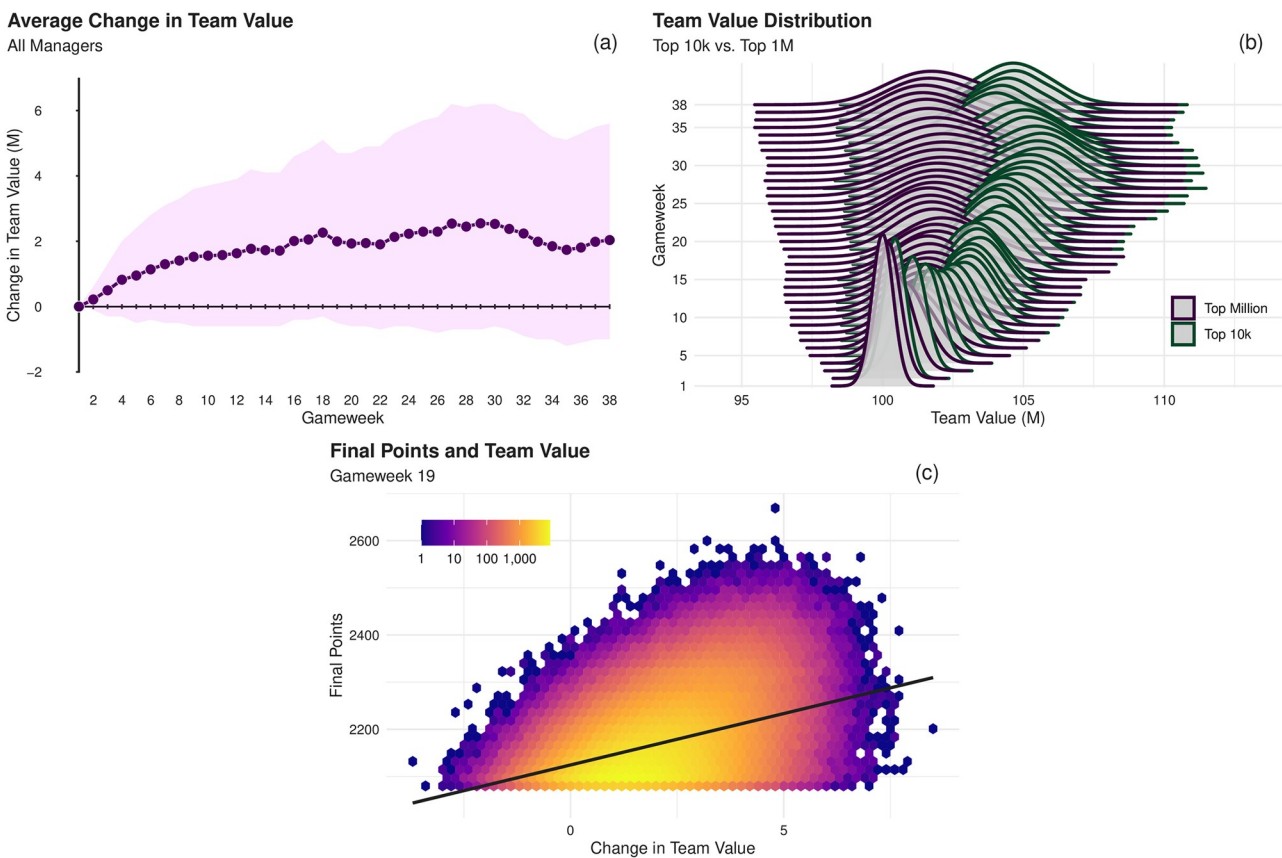

**Fig 4. Analysis of the team value of managers.** (a) The change in average team value from the initial £100M of all managers, along with 95 percentiles; note the general upward trend of team value over the course of the season. (b) Distributions of team values for each gameweek for those who finished in the top ten thousand positions (i.e., the combination of those in the top $10^3$ and $10^4$ tiers) versus lower-ranked managers. The distribution for those with higher rank is generally to the right of that describing the other managers from an early stage of the season, indicating higher team value being a priority for successful managers. (c) The relationship between a manager's team value at GW 19 versus their final points total, where the heat map indicates the number of managers within a given bin. The black line indicates the fitted linear regression line, showing that an increase in team value by £1M at this point in the season results in an average final points increase of 21.8 points.

performance, by providing additional opportunities to obtain points. Said game-chips offer specific opportunities for managers to take advantage of numerous scenarios which occur within a season that offer the potential for obtaining players with a higher likelihood of gaining additional points. These include the possibility of certain players appearing twice in one week thus having two matches to gain points (or conversely weeks in which they do not feature at all) or a team appearing in a highly favourable fixture. The time at which these chips are played and the corresponding points obtained are one observable element of a managers' strategy. A detailed description for each of the chips and analysis of the approach taken by the managers in using them is given in S5 Note in S1 File.

For the sake of brevity we focus here only on one specific chip, the *bench boost*. When this chip is played, the manager receives points from all fifteen players in their squad in that GW, rather than only the starting eleven as is customary. This clearly offers the potential for a large upswing in points if this chip is played in an efficient manner, and as such it should ideally be used in GWs where the manager may otherwise struggle to earn points with their current team or weeks in which many of their players have a good opportunity of returning large point scores. The double and blank GWs might naively appear to be optimal times to deploy this

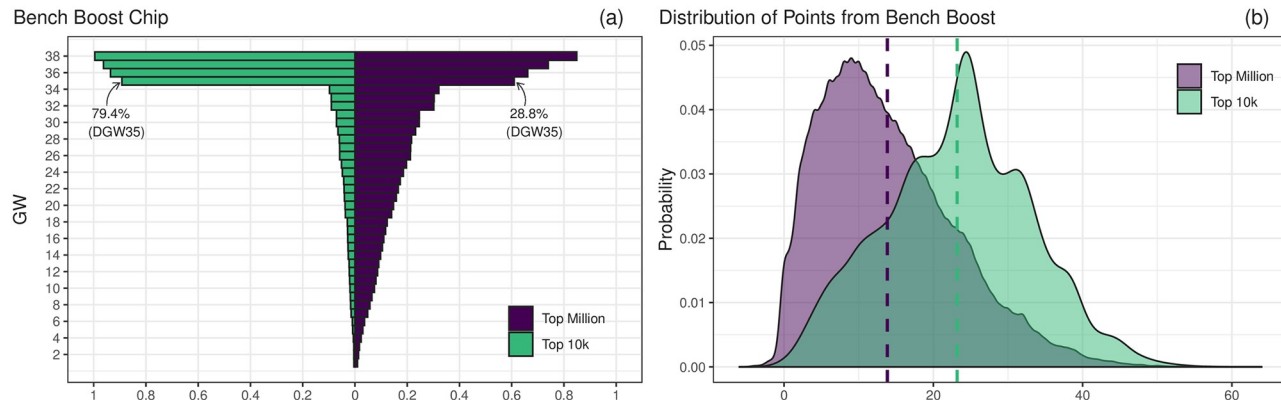

**Fig 5. Summary of use and point returns of the bench boost chip.** The managers are grouped into two groups: those who finished in the top ten-thousand positions (Top 10k) and the remainder (Top Million). (a) Fraction of managers who had used the bench boost chip by each gameweek. We see a clear strategy for use in double gameweek 35, particularly for the top managers, 79.4% of whom used it at this stage. (b) Distribution of points earned from using this chip along with the average points—23.2 for the Top 10k and 13.8 for the Top Million—shown by the dashed lines.

chip however when the managers' actions are analysed we see differing approaches (and corresponding returns).

Fig 5 shows the proportion of managers who had used the bench boost chip by each GW alongside the corresponding distribution of points the manager received from this choice, where we have grouped the two higher tiers into one group and the remaining managers in another for visualization purposes (see S10 and S11 Figs in S1 File, and S7-S10 Tables in S1 File for a breakdown of use and point returns by each tier). It is clear that the majority of better performing managers generally focused on using these chips during the double and blank GWs with 79.4% choosing to play their BB chip during DGW35 in comparison to only 28.9% of those in the rest of the dataset. We also observe the difference in point returns as a result of playing the chip, with the distribution for the top managers being centred around considerable higher values, demonstrating that their squads were better prepared to take advantage of this chip. The fact that the managers were willing to wait until one of the final gameweeks is also indicative of the long-term planning that separates them from those lower ranked. Similar results can be observed for the other game-chips (S8-S10 Tables in S1 File). We also highlight that a large proportion of managers made use of other chips in GW36, which was the later gameweek in which there was a large fluctuation from the average shown in Fig 2.

Finally, we comment on the fact that some managers did not employ their chips by the game's conclusion which suggests that either they were not aware of them or, more likely, the mangers in question had simply lost interest in the game at this point. As such, the quantity of managers who had not used their chip gives us a naive estimation of the retention rate for active managers in Fantasy Premier League (85.05% of managers in our dataset). We note that this is a biased estimate in the sense that our dataset is only considering the top tiers of managers, or at least those who finished in the top tiers, and one would expect the drop out rate to be in fact much higher in lower bands.

## Template team

While the preceding analysis proposes reasons for the differences between points obtained by tiers shown in Fig 2, the question remains as to why the managers' gameweek points totals show similar temporal dynamics. In order to understand this we consider here the underlying structure of the managers' teams. We show that a majority of teams feature a core group of

players that results in a large proportion of teams having a similar make-up. We call this phenomenon the *template team* which appears to emerge at different points in the season; this type of collective behaviour has been observed in such social settings previously, see, for example [38, 39]. We identify the template team by using the network structure describing the teams of all managers, which is described by the adjacency matrix $A_{ij}^G$, whereby an edge between two players $i$ and $j$ appearing in $n$ teams for a given gameweek $G$ describes a value in the matrix given by $A_{ij}^G = n$. This matrix is similar in nature to the co-citation matrix used within the field of bibliometrics [40], see Fig 6 for a representation of the process.

With these structures in place we proceed to perform hierarchical *k*-means clustering on the matrices in order to identify groups of players constituting the common building blocks of the managers' teams. By performing the algorithm with *k* = 4 clusters, with this number of clusters being identified using the elbow method (see Methods), we find that three of said clusters contain only a small number of the 624 players, suggesting that most teams include this small group of core players (see S6 Table in S1 File for the identities of those in the first cluster each gameweek). Fig 7(a) shows the size of these first three clusters over all managers for each gameweek of the season (S8 Fig in S1 File shows the equivalent values for each tier). To understand this result further, consider that at their largest these three clusters only consist of 5.13% (32/624) of the available players in the game, highlighting that the teams are congregated

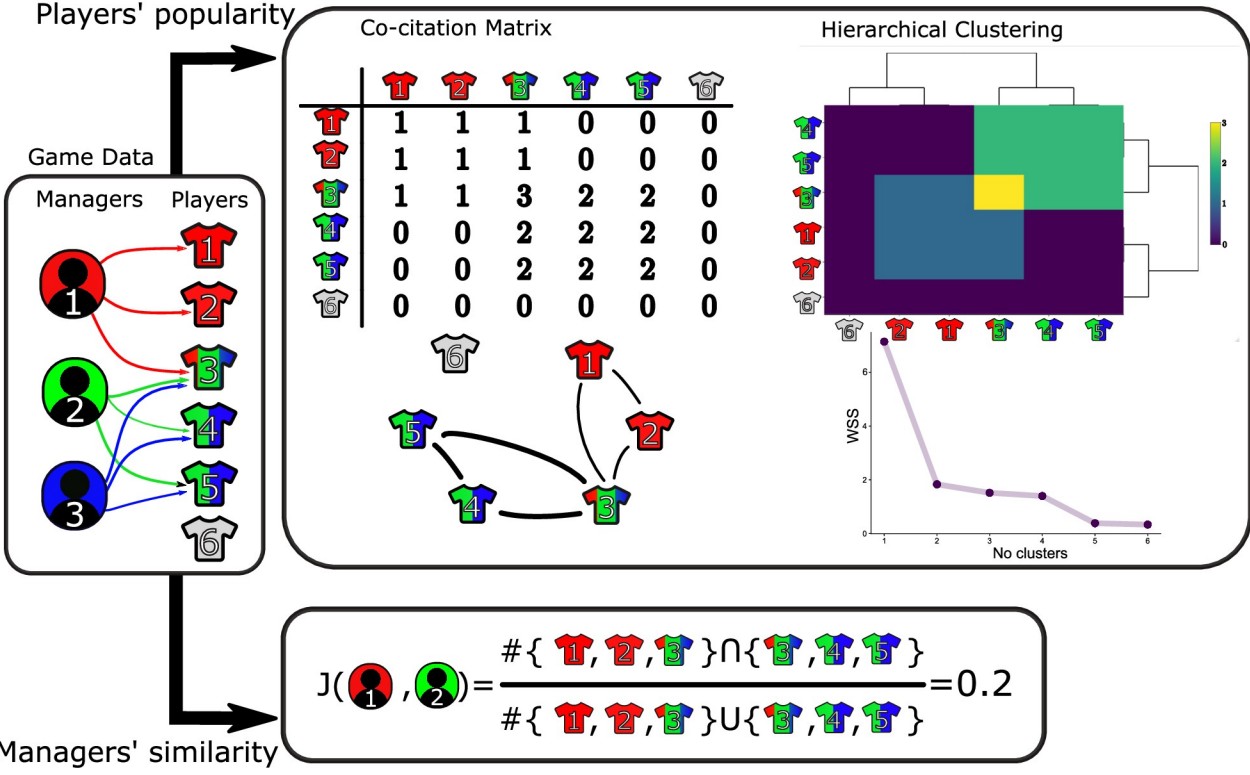

**Fig 6. Schematic representation of the approaches taken to identify similarity between the composition of managers' teams in each GW.** We view the connections between managers and players as a bipartite network such that an edge exists if the player is in the managers' team. To determine the relationship between players' levels of popularity we use the co-occurrence matrix which has entries corresponding to the number of teams in which two players co-appear. Using this matrix we perform hierarchical clustering techniques to identify groups of players who are similarly popular within the game, where the number of clusters is determined by analysing the within-cluster sum of squared errors. The similarity between the teams of two managers is determined by calculating the Jaccard similarity, which is determined by the number of players that appear in both teams.

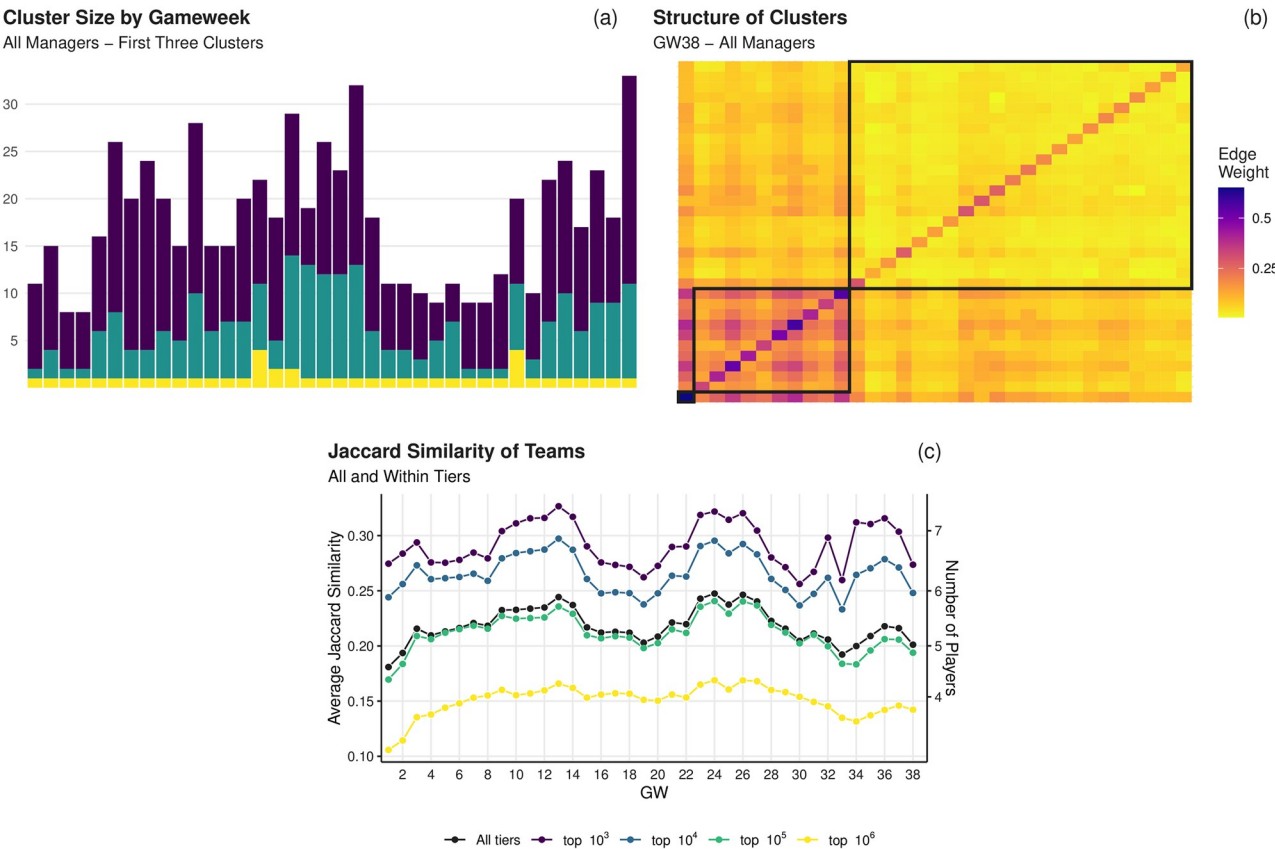

**Fig 7. Analysis of team similarity of managers.** (a) Size of each of the first three identified clusters over all managers for each gameweek. Note that the first cluster is generally of size one, simply containing the most-owned player in the game. (b) An example of the network structure of these three clusters for gameweek 38, where we can see the ownership level decreasing in the larger clusters. The diagonal elements of this structure are the fraction of teams in which the player is present. (c) The Jaccard similarity between the tiers of managers and also over all managers; note that the higher-performing managers tend to be more like one another than those in lower tiers, also note the fluctuations in similarity over the course of the season indicating that a template team emerges at different time points.

around a small group of players. For an example representation of this matrix alongside its constituent clusters we show the structure in panel (b) of Fig 7 for gameweek 38, which was the point in time at which the three clusters were largest.

To further examine the closeness between managers' decisions we consider the Jaccard similarity between sets of teams, which is a distance measure that considers both the overlap and also total size of the sets for comparison (see Methods for details). Fig 7(c) shows the average of this measure over pairwise combinations of managers from all tiers and also between pairs of managers who are in the same tier.

Fluctuations in the level of similarity over the course of the season can be seen among all tiers indicating times at which teams become closer to a template followed by periods in which managers appear to differentiate themselves more from the peers. Also note that the level of similarity between tiers increases with rank suggesting that as we start to consider higher performing managers, their teams are more like one another not only at certain parts of the season but, on average, over its entirety (see S9 Table in S1 File for corresponding plots for each tier individually). The high level of similarity between the better managers' teams in the first gameweek (and the corresponding large points totals seen in S1 Fig in S1 File) is particularly interesting given that this is before they have observed a physical game being played in the actual

season. This suggests a similar approach in identifying players based purely upon their historical performance and corresponding value by the more skilled managers.

## Discussion

The increasing popularity of fantasy sports in recent years [27] enables the quantitative analysis of managers' decision-making through the study of their digital traces. The analysis we present in this article considers the game of Fantasy Premier League, which is played by approximately seven million managers. We observe a consistent level of skill among managers in the sense that there exists a considerable correlation between their performance over multiple seasons of the game, in some cases over thirteen years. This result is particularly striking given the stochastic nature of the underlying game upon which it is based.

Encouraged by these findings, we proceeded to conduct a deeper analysis of the actions taken by a large proportion of the top one million managers from the 2018-19 season of the game. This allowed each decision made by these managers to be analysed using a variety of statistical and graphical tools. We divided the managers into tiers based upon their final position in the game and observed that the managers in the upper echelons consistently outperformed those in lower ones, suggesting that their skill levels are present throughout the season and that their corresponding rank is not dependent on just a small number of events. The skill-based decisions were apparent in all facets of the game, including making good use of transfers, strong financial awareness, and taking advantage of short- and long-term strategic opportunities, such as their choice of captaincy and use of the chips mechanic.

Arguably the most remarkable observation presented in this article is, however, the emergence of what we coin a *template team* that suggests a form of common collective behaviour occurring between managers. We show that most teams feature a common core group of constituent players at multiple time points in the season. This occurs despite the wide range of possible options for each decision, suggesting that the managers are acting similarly, and particularly so for the top-tier managers as evident by their higher similarity metrics. Such coordinated behaviour by managers suggests an occurrence of the so-called 'superstar effect' within fantasy sports just as per their physical equivalent [41], whereby managers independently arrive at a common conclusion on a core group of players who are viewed as crucial to optimal play. A further dimension is added by the fact that the similarity between the teams of better managers is evident even prior to the first event of the season, i.e., they had apparently all made similar (good) decisions even 'before a ball was kicked'.

In this article we have focussed on the behaviour of the managers and their decision-making that constitutes their skill levels. The availability of such detailed data offers the potential for further research from a wide range of areas within the field of computational social science. For example, analysis of the complex financial dynamics taking place within the game as a result of the changing player values and the buying/selling decisions made by the managers would be interesting. A second complementary area of research would be the development of algorithms that consider the range of possible options available to managers and give advice on optimizing point returns. Initial analysis has recently been conducted [33] in this area, including the optimal captaincy choice in a given gameweek, and has demonstrated promising results.

In summary, we believe the results presented here offer an insight into the behaviour of top fantasy sport managers that is indicative of both long-term planning and collective behaviour within their peer group, demonstrating the intrinsic level of skill required to remain among the top positions over several seasons, as observed in this study. We are however aware that the correlations between decisions and corresponding points demonstrated are not perfect,

which is in some sense to be expected due to the non-deterministic nature which makes the sport upon which the game is based so interesting to the millions of individuals who enjoy it each week. These outcomes suggest a combination of skill and luck being present in fantasy sport just as in their physical equivalent.

## Methods

### Data collection

We obtained the data used in this study by accessing approximately 50 million unique URLs through the Fantasy Premier League API. The rankings at the end of the 2018/19 season were obtained through https://fantasy.premierleague.com/api/leagues-classic/league-id/standings/ from which we could obtain the entry ID of the top 1 million ranked managers. Using these IDs we then proceeded to obtain the team selections along with other manager quantities for each gameweek of this season that were used in the study through https://fantasy. premierleague.com/api/entry/entry-id/event/GW/picks/, we then filtered the data to include only managers for whom we had data for the entirety of the season which consisted of 901, 912 unique managers. The data for individual footballers and their performances were captured via https://fantasy.premierleague.com/api/bootstrap-static/. Finally, the historical performance data was obtained for 6 million active managers through https://fantasy.premierleague.com/api/entry/entry-id/history/.

### Calculation of transfer quality

In order to calculate the transfer quality plot shown in Fig 3(b) we consider the gameweeks in which managers made one transfer and, based upon the value of the player whom they transferred in, determine what fraction of players with the same price or lower the manager could have instead bought for their team. Suppose that in gameweek $G$ the manager transferred out player $x_i$, who had value $q_G(x_i)$, for player $x_j$ who scored $p_G(x_j)$ points in the corresponding gameweek. The calculation involves firstly finding all players the manager could have transferred in, i.e., those with price less than or equal to $q_G(x_i)$ and then determining the fraction $y_G(x_i, x_j)$ of these players who scored more points than the chosen player given the player whom was transferred out. This is calculated by using

$$y_G(x_i, x_j) = \frac{\sum_k \mathbb{1}[q_G(x_k) \leq q_G(x_i)] \cdot \mathbb{1}[p_G(x_k) > p_G(x_j)]}{\sum_\ell \mathbb{1}[q_G(x_\ell) \leq q_G(x_i)]},$$

where $\mathbb{1}$ represents the indicator function. Using this quantity we proceed to group over the entire season for each tier of manager which allows us to obtain the distribution of the measure itself and finally the probability of making a better transfer which is shown in panel (b) of Fig 3.

### Team similarity

With the aim of identifying levels of similarity between the teams of two managers $i$ and $j$ we make use of the Jaccard similarity which is a measure used to describe the overlap between two sets. Denoting by $T_i^G$ the set of players that appeared in the squad of manager $i$ during gameweek $G$ we consider the Jaccard similarity between the teams of managers $i$ and $j$ for gameweek $G$ given by

$$J^G(i, j) = \frac{|T_i^G \cap T_j^G|}{|T_i^G \cup T_j^G|},$$

where $|\cdot|$ represents the cardinality of the set. We then proceed to calculate this measure for all $n$ managers which results in a $n \times n$ symmetric matrix $J^G$, the $(i, j)$ element of which is given by the above equation, note that the diagonal elements of this matrix are unity. Calculation of this quantity over all teams is computationally expensive in the sense that one must perform pairwise comparison of the $n$ teams for each gameweek. As such we instead calculated an estimate of this quantity by taking random samples without replacement of 100 teams from each tier and calculating the measure both over all teams and also within tiers for each gameweek. We repeat this calculation 10,000 times and the average results are those used in the main text and S4 Note in S1 File.

## Cluster identification of player ownership

As described in the main text, the calculation of clusters within which groups of players co-appear involves taking advantage of the underlying network structure of all sets of teams. The adjacency matrix describing this network is defined by the matrix $A_{ij}^G$ that has entry $(i, j)$ equal to the number of teams within which player $i$ and $j$ co-appear in gameweek $G$. Note that the diagonal entries of this matrix describe the number of teams in which a given player appears gameweek $G$. Using this matrix we identify the clusters via a hierarchical clustering approach. Specifically, we implement *k-means* clustering and determine an appropriate number of clusters by considering the within-cluster sum of squared errors for a range of possible values before identifying the number of clusters at which the rate this error decreases slows down using the elbow method (see S4 Note in S1 File for further information on this approach). Through this method we find $k = 4$ clusters to be an appropriate value and is used throughout the analysis presented in the text.

## Supporting information

**S1 File. The supporting information file includes S1–S11 Figs and S1–S8 Tables, in addition to some additional explanation and discussion.**
(PDF)

## Acknowledgments

Helpful discussions with Kevin Burke, James Fannon, Peter Grinrod, Stephen Kinsella, Renaud Lambiotte, and Sean McBrearty are gratefully acknowledged. We acknowledge the DJEI/DES/SFI/HEA Irish Centre for High-End Computing (ICHEC) for the provision of computational facilities and support.

## Author Contributions

**Conceptualization:** Joseph D. O'Brien, James P. Gleeson, David J. P. O'Sullivan.

**Data curation:** Joseph D. O'Brien.

**Formal analysis:** Joseph D. O'Brien, David J. P. O'Sullivan.

**Funding acquisition:** James P. Gleeson.

**Investigation:** Joseph D. O'Brien, David J. P. O'Sullivan.

**Methodology:** Joseph D. O'Brien, David J. P. O'Sullivan.

**Project administration:** Joseph D. O'Brien.

**Supervision:** James P. Gleeson.

**Visualization:** Joseph D. O'Brien.

**Writing – original draft:** Joseph D. O'Brien.

**Writing – review & editing:** Joseph D. O'Brien, James P. Gleeson, David J. P. O'Sullivan.

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
