## [Decision Letter · Decision Letter 0]

13 Jan 2021

PONE-D-20-32476

Identification of skill in an online game: The case of Fantasy Premier League

PLOS ONE

Dear Dr. O'Brien,

Thank you for submitting your manuscript to PLOS ONE. After careful consideration, we feel that it has merit but does not fully meet PLOS ONE’s publication criteria as it currently stands. Therefore, we invite you to submit a revised version of the manuscript that addresses the points raised during the review process.

We have received just one review. Because I agree with Reviewer #1 in that this is a well-written paper that requires minor revision before it can be accepted for publication, I decided not to further delay the peer-review process. If you decide to resubmit a revised version, please provide point-to-point responses to each of the comments raised by the reviewer. 

We look forward to receiving your revised manuscript.

Kind regards,

Anthony C Constantinou

Academic Editor

PLOS ONE

Journal Requirements:

Reviewers' comments:

Reviewer's Responses to Questions

**Comments to the Author**

1. Is the manuscript technically sound, and do the data support the conclusions?

Reviewer #1: Yes

2. Has the statistical analysis been performed appropriately and rigorously? 

Reviewer #1: Yes

3. Have the authors made all data underlying the findings in their manuscript fully available?

Reviewer #1: Yes

4. Is the manuscript presented in an intelligible fashion and written in standard English?

Reviewer #1: Yes

5. Review Comments to the Author

Reviewer #1: Very well written paper. I only have minor recommendations, so I recommended publish.

First their statistical performances. It is unclear whether you are talking about players or managers in fantasy.

Besides reference 28. A study performed by Easton and Newell in the Journal of Sports Analytics showed that chance doesn't win daily fantasy sports and should be added.

Remove the here before introduce in the second paragraph.

The sentence that starts In section IIC3... You may want to add a comment as to why. That is a player plays in more games for the week and thus will earn more points than if the player played in fewer games that week. You should state the claim better than I did though.

With these structures in place....

You comment on hierarchial clustering. I would be more specific. In the details you comment about elbow method and K-means. I would bring these methods into the main body of the paper in all instances. The readers must know the tools you used. Then you can point them to more details in the supplementary sections.

Nice job

6. PLOS authors have the option to publish the peer review history of their article (what does this mean?). If published, this will include your full peer review and any attached files.

Reviewer #1: No

---

## [Author Response · Author response to Decision Letter 0]

19 Jan 2021

Dear Prof. Constantinou,

We thank you and the reviewer for the careful reviews and the positive and constructive comments on our manuscript. We hope that the revised manuscript, with changes in blue text, will be considered worthy of publication in the PLOS ONE. 

Yours sincerely,

Joseph O’Brien (on behalf of all authors).

Reviewer #1: Very well written paper. I only have minor recommendations, so I recommended publish.

We thank the Reviewer for their extremely positive feedback on our manuscript and recommendation for publication. The reviewer highlights areas in which the clarity of the text could be enhanced: A number of textual edits have been implemented throughout and these changes have resulted in the clarity of the work being considerably improved.

First their statistical performances. It is unclear whether you are talking about players or managers in fantasy.

We appreciate that there may be some confusion due to the managers ‘playing’ the game which itself is based on the performance of football players. A statement is now included in the introduction which specifically sets out our aim of a statistical analysis of manager performance (rather than players) and we feel this should set the reader on a clearer track for the remainder of the manuscript.

Besides reference 28. A study performed by Easton and Newell in the Journal of Sports Analytics showed that chance doesn't win daily fantasy sports and should be added.

We thank the reviewer for pointing out this reference to us and it has now been included in conjunction with reference 28 to highlight the emerging research into fantasy sports that shows these games involve considerable levels of skill.

Remove the here before introduce in the second paragraph.

Implemented as requested.

The sentence that starts In section IIC3... You may want to add a comment as to why. That is a player plays in more games for the week and thus will earn more points than if the player played in fewer games that week. You should state the claim better than I did though.

We agree with the reviewer that examples of how managers may take advantage of these players appearing multiple times would help the reader and a commentary has been included in the revised manuscript.

With these structures in place....

You comment on hierarchical clustering. I would be more specific. In the details you comment about elbow method and K-means. I would bring these methods into the main body of the paper in all instances. The readers must know the tools you used. Then you can point them to more details in the supplementary sections.

We agree with the reviewer that further commentary on the clustering approaches used in our analysis would help the reader. As such the choice of k-means and the identification of the number of clusters via the elbow method is now explicitly mentioned in Section 3D, and furthermore we now provide more detailed commentary on the approach in the main text rather than in supplemental material (Sec 4D).

---

## [Editor Report · Decision Letter 1]

25 Jan 2021

Identification of skill in an online game: The case of Fantasy Premier League

PONE-D-20-32476R1

Dear Dr. O'Brien,

We’re pleased to inform you that your manuscript has been judged scientifically suitable for publication and will be formally accepted for publication once it meets all outstanding technical requirements.

Kind regards,

Anthony C Constantinou

Academic Editor

PLOS ONE
---

## [Editor Report · Acceptance letter]

8 Feb 2021

PONE-D-20-32476R1 

Identification of skill in an online game: The case of Fantasy Premier League 

Dear Dr. O'Brien:

I'm pleased to inform you that your manuscript has been deemed suitable for publication in PLOS ONE. Congratulations! Your manuscript is now with our production department. 

Kind regards, 

on behalf of

Dr. Anthony C Constantinou 

Academic Editor

PLOS ONE